# Oxidative Damage and Post-COVID Syndrome: A Cross-Sectional Study in a Cohort of Italian Workers

**DOI:** 10.3390/ijms24087445

**Published:** 2023-04-18

**Authors:** Angela Stufano, Camilla Isgrò, Luigi Leonardo Palese, Paolo Caretta, Luigi De Maria, Piero Lovreglio, Anna Maria Sardanelli

**Affiliations:** 1Interdisciplinary Department of Medicine, University of Bari Aldo Moro, Piazza G. Cesare 11, 70124 Bari, Italy; 2Department of Translational Biomedicine Neuroscience, University of Bari Aldo Moro, Piazza G. Cesare 11, 70124 Bari, Italy

**Keywords:** long COVID, oxidative stress, biomarker, malondialdehyde, workers, TBARS assay

## Abstract

In addition to the acute symptoms after infection, patients and society are also being challenged by the long-term effects of COVID-19, known as long COVID. Oxidative stress, as a pivotal point in the pathophysiology of COVID-19, could potentially be also involved in the development of the post-COVID syndrome. The aim of the present study was to evaluate the relationship between changes in oxidative status and the persistence of long-COVID symptoms in workers with a previous mild COVID-19 infection. A cross-sectional study was conducted among 127 employees of an Italian university (80 with a previous COVID-19 infection, and 47 healthy subjects). The TBARS assay was used to detect malondialdehyde serum levels (MDA), while total hydroperoxide (TH) production was measured by a d-ROMs kit. A significant difference in mean serum MDA values was found between previously infected subjects and healthy controls and (4.9 µm vs. 2.8 µm, respectively). Receiver–operating characteristic (ROC) curves showed high specificity and good sensibility (78.7% and 67.5%, respectively) for MDA serum levels. A random forest classifier identified the hematocrit value, MDA serum levels, and IgG titer against SARS-CoV-2 as features with the highest predictive value in distinguishing 34 long-COVID from 46 asymptomatic post-COVID subjects. Oxidative damage persists in subjects with previous COVID-19 infection, suggesting a possible role of oxidative stress mediators in the pathogenesis of long COVID.

## 1. Introduction

Coronavirus disease 2019 (COVID-19), a highly transmissible disease caused by the severe acute respiratory syndrome coronavirus 2 (SARS-CoV-2), has presented extraordinary challenges to the healthcare systems and occupational settings. Although the vast majority of the infected subjects survive, the impact of COVID-19 remains urgent due to the wide range of outcomes which can emerge following the acute phase of the illness [1]. This post-acute syndrome referred to as long COVID (or post-acute sequelae of COVID or post-acute COVID-19 syndrome), represents a significant challenge for patients, physicians, and economic systems because the causes, patient management, and even symptom patterns remain difficult to characterize [2]. Long COVID is usually defined as “signs and symptoms developed during or following a disease consistent with COVID-19, and which continue for more than four weeks but they are not explained by alternative diagnoses” [3,4]. Since the persistent effects of COVID-19 were recognized 6 months into the pandemic, a variety of clinical presentations, with up to 200 symptoms, and degrees of severity have been reported. There is still uncertainty about the true prevalence of long COVID, because the diagnosis is based only on clinical symptoms, previous COVID-19 infection, and the lack of an alternative cause [5]. Although this condition is mostly reported in severe and critical disease survivors, the lasting effects also occur in asymptomatic individuals or individuals with mild infections who did not require hospitalization. Additionally, it has not yet been definitely established if gender, age, ethnicity, job, underlying health conditions, and viral load significantly affect the risk of developing long-term effects of COVID-19 [6].

Developing the knowledge on host factors that predict a long-hauler status, as well as potential association with symptom clusters, will be pivotal to understanding why the sequelae of acute COVID-19 infection vary widely from patient to patient, from complete recovery to severe persistent symptoms affecting multiple organs and mental health, and for the consequent development of evidence-based management guidelines [7]. Little is known about the mid- and long-term consequences of COVID-19 in non-hospitalized individuals, although emerging data suggest that a significant proportion of mild or moderate cases, even in younger adults, have persistent symptoms associated with previous SARS-CoV-2 infection [8]. It is still unclear, however, whether long COVID reflects the tissue persistence of the virus or is promoted by an aberrant inflammatory autoimmune response or by the primary organ damage triggered by the acute infection [9]. It has also been suggested that cellular damage, a robust innate immune response with inflammatory cytokine production, and a procoagulant state induced by SARS-CoV-2 infection may contribute to the pathogenesis of long COVID [10].

Oxidative stress is a natural process that occurs during metabolism and plays an important role in maintaining balance in prooxidant–antioxidant levels in cells, tissue, and organs [11]. During abnormal physiological conditions, uncontrolled production of highly reactive oxygen species (ROS) occurs, promoting a cascade of biological events inducing pathological host responses [12]. ROS attack various classes of biomolecules, including proteins, DNA, and lipids such as polyunsaturated fatty acids (PUFAs). Particularly, the PUFA arachidonic acid is peroxidized to form malondialdehyde (MDA), 4-hydroxy-2-nonenal (HNE), and other reaction products, such as F2-isoprostanes, widely accepted as biomarkers of oxidative damage, namely, lipid peroxidation [13]. Disturbances in redox balance have been observed in different inflammatory and viral diseases. Viruses generally lead to disturbance of redox homeostasis in infected cells and increased production of ROS in activated phagocytes [14]. Efforts to decipher the pathophysiology of SARS-CoV-2 infection, therefore, have also focused on the role of free radicals in mechanisms, such as the interaction with the receptor, viral replication, the interplay between the cytokine storm and the free radical storm in the virus-induced hyperactive immune response, and the subsequent end-organ damage [15]. It has been suggested that endothelial dysfunction induced by impaired hypoxia-related redox signaling may also contribute to SARS-CoV-2 pathogenicity [16]. However, to date, no data are available regarding any change in oxidative stress biomarkers in relation to long COVID syndrome. The purpose of the present study has been to evaluate the possible relationship between the presence of any change in the levels of oxidative stress and lipid peroxidation biomarkers and the persistence of long COVID symptoms at least four months after the acute infection in workers with previous mild COVID-19, not requiring hospitalization.

## 2. Results

Table 1 summarizes the general characteristics and clinical parameters of the workers with previous SARS-CoV-2 infection (COVID-19 patients) at four months after negativization, and of the controls. A significant difference was found only in alcohol consumption, which was higher in the controls than in COVID-19 patients (mean value 3.6 ± 4.6, range 0–14 vs. mean value 2.1 ± 5.5, range 1–7 alcoholic units per week).

Moreover, significant differences were found in the red blood cell, white blood cell, and platelet counts, and hematocrit values, which were significantly lower in the COVID-19 patients than in the controls, and in the average values of mean corpuscular volume (MCV) and mean corpuscular hemoglobin concentration (MCHC), which were significantly lower in the controls. Finally, a significant difference in serum MDA values was found to be higher in COVID-19 patients than in the controls (mean value 4.9 µm vs. 2.8 µm) (Table 1).

In order to find a possible predictive index of post-COVID-19 status, we performed receiver–operating characteristic (ROC) curve analyses of H_2_O_2_ mg/dL and MDA µm (Figure 1). The ROC curve of H_2_O_2_ presented a low area under the curve (AUC) proving that this index is not able to discriminate COVID-19 patients from healthy controls (Figure 1a). The ROC curve of MDA was otherwise characterized by good specificity and sensibility (78.7% and 67.5%, respectively), identifying MDA as a good predictive index of COVID-19 patients, with a cut-off value of 3.3 µm and AUC of 76.3% (Figure 1b).

Thirty-four long-COVID (LC) patients were identified among the group of COVID-19 patients, selecting those who reported the persistence of at least one symptom after infection, or the appearance of symptoms that could not be explained by another cause and were plausibly related to COVID-19 infection. The MDA levels significantly differ in LC patients when compared with both the controls (*p* < 0.001) and non-long-COVID (nLC) patients (*p* < 0.001), while no difference was found between nLC patients and the controls (Figure 2).

Demographic characteristics and clinical variables of COVID-19 patients subdivided according to the presence of symptoms (LC vs. nLC patients, respectively) are shown in Table 2. A higher prevalence of the female sex and patients requiring oxygen therapy during the acute infection was found in the LC than in nLC patients (19% vs. 6%, and 4% vs. 0%, respectively, *p* < 0.001). In addition, this group had significantly higher IgG values and lower mean hematocrit values than the nLC patients (Table 2).

No significant differences between the LC and nLC patients were identified in the duration of symptoms in the acute phase of infection (Table 3), nor in the duration and prevalence of specific symptoms during the acute phase, except for cough (20% vs. 16%, mean duration 9.2 vs. 3.1 days, *p* < 0.001) and headache (mean duration 2.5 vs. 0.7 days, *p* < 0.05).

Twenty different symptoms were found in the LC patients, with a higher frequency of asthenia, insomnia, and peripheral paresthesia, followed by concentration difficulties (Table 3).

To confirm the role of some parameters as long-COVID predictors, we applied a random forest classifier to identify the features that have the highest predictive value in distinguishing LC from nLC patients (Figure 3). The variables that showed the most significance in distinguishing the two groups of subjects were found to be the hematocrit value, MDA value, and specific IgG titer against SARS-CoV-2.

## 3. Discussion

In this study, the association between COVID-19 and peroxides and MDA serum concentrations as biomarkers for systemic oxidative stress was examined in a cohort of non-hospitalized patients four months after negativization. Most notably, our findings revealed that serum MDA was significantly higher in COVID-19 patients as compared with healthy controls, providing valuable evidence of the presence of a systemic redox imbalance four months after the SARS-CoV-2 infection, even in relatively milder cases, not requiring hospital admission.

Several studies suggested an association between oxidative stress and COVID-19 pathogenesis, while little data are available related to the relationship between long COVID and oxidative damage [16,17]. A significant correlation between oxidative stress markers and respiratory viral infections has been previously demonstrated, particularly for RNA viruses [18], such as HCV [19], HIV [20], Zika [21], H1N1, and influenza [22], which have been shown to promote an elevated level of oxidative stress. Oxidative stress by virus infection is also linked to the severity of the disease; it is related to the activation of innate immunity by cytokine production and involved in the facilitation of virus replication inside the cell [23].

Recent studies have demonstrated that the overproduction of ROS could have a major role in the pathogenesis, progression, and severity of COVID-19, showing that acute infection by SARS-CoV-2 is accompanied by increased expression of ROS-responses and glycolysis in peripheral blood mononuclear cells, increased MDA, 8-hydroxy-2′-deoxyguanosine, and nitrosine levels [24,25]. Several mechanisms have been related to increased oxidative damage during COVID-19 infection. First, the ACE receptor for SARS-CoV-2 is downregulated by binding to the virus, leading to an increased presence of superoxide species because angiotensin II degradation into angiotensin 1-7 mitigates oxidative stress as it inhibits NADPH oxidase-mediated ROS production [26]. Moreover, during SARS-CoV-2 infection, the NADPH oxidase complex is activated by the chemotaxis of macrophages and neutrophils, which leads to ROS production, including superoxide radical anion and hydrogen peroxide production [27]. Innate immunity activates transcription factors, such as NF-ƙB, which promotes the overexpression of pro-inflammatory genes, resulting in an exacerbated pro-inflammatory host response [28]. Disseminated intravascular coagulopathy, sepsis, and reduced oxygen transport to the tissues are shown in COVID-19 patients [29]. Hypoxia can produce reactive species, such as superoxide and H_2_O_2_, which can upregulate the expression of inflammatory cytokines. In turn, inflammatory cytokines can increase oxidative stress markers via the activation of macrophages, neutrophils, and endothelium cells, creating a vicious cycle of oxidative damage and inflammation [30].

This interaction between oxidative stress and inflammatory cytokines can lead to several organ failures in COVID-19 patients and might potentially have a role in the development of long COVID. Since increased free radical production, as a hallmark of SARS-CoV-2 infection, has the capacity to modify macromolecules over time, this cumulative oxidative damage may contribute to many mechanisms underlying COVID-19 progression.

High serum MDA levels were observed in our COVID-19 patients compared with the healthy controls. MDA levels have been shown to predict worse clinical outcome in previous studies, including patients with cardiovascular diseases, Alzheimer [31], multiple sclerosis [32], and COPD [33], and they are commonly considered a marker of ferroptosis, a form of regulated cell death characterized by iron-dependent lipid peroxidation, which induces cell death [34]. Lipid peroxidation affects all cell membranes, inducing damage and loss of function, and is involved in several disease conditions, playing a role in both homeostasis and response to stress, such as viral infections [35].

Our results seem to be in line with previous reports, which demonstrate elevated levels of MDA in COVID-19 patients, suggesting overproduction of free radicals during the infection, which in turn destroy lipid membranes [36,37,38]. However, to the best of our knowledge, this seems to be one of the first reports underlying the persistence of oxidative damage four months after the acute COVID-19 infection, even in subjects with asymptomatic or mild infections. As such, our work demonstrates a persistent imbalance between oxidative stress and toxicity, which may have a variety of detrimental effects.

Moreover, our analysis shows that elevated serum level of MDA seems to be related to the presence of symptoms four months after COVID-19, suggesting that the persistence of oxidative damage could be a predictive factor for the development of the long-COVID syndrome. According to this, a recent study on subjects with long-COVID syndrome demonstrated that around 60% of the variance in neuropsychiatric symptoms was explained by an imbalance between prooxidant and antioxidant factors, suggesting that the impact of acute COVID-19 on the symptoms of long COVID could be partly mediated by a lipid peroxidation–associated aldehyde formation [39].

In our study, after analyzing different variables according to sex, long COVID was prevalent in the female sex and in subjects with a higher mean IgG antibody titer against SARS-CoV-2 and with lower hematocrit mean values; the random forest classifier identified the hematocrit value, the MDA serum level, and the IgG titer against SARS-CoV-2 as the variables most significant in distinguishing LC from nLC patients.

Several previous studies, as observed in our study, showed a higher prevalence of the long-COVID syndrome in the female sex [40,41]. The potential mechanisms hypothesized for this difference included a disturbance of physiological ovarian steroid hormone production following COVID-19 and an altered chronic inflammatory response due to sex-based immunomodulation, suggesting the presence of common immune dysregulation pathways related to the higher prevalence of immune diseases in the female sex. According to our results, showing significantly higher mean SARS-CoV-2 IgG titers in LC patients, immunological dysfunctions have been detected in different studies including long-COVID subjects. These studies have shown that LC patients had highly activated innate immune cells, elevated expression of type I IFN and type III IFN, with higher production of a pro-inflammatory cytokine such as IL-6, indicating that components of the acute inflammatory response are associated with long-COVID [42,43,44]. It is possible to infer that systemic inflammation is bidirectionally linked to oxidative damage in long-COVID; persistent inflammation generates ROS and lipid peroxidation, and redox imbalance causes cellular damage that evokes an inflammatory response, leading to a vicious cycle [45].

Our study found no difference in the serum concentration of H_2_O_2,_ which is not consistent with MDA serum levels. This could be due to the shorter half-life of this compound compared with MDA, which is a more stable end product of lipid peroxidation [46]. Similarly, a previous report demonstrates an increase in serum MDA concentration in COVID-19 patients 14 days after hospital admission, suggesting a similar time course pattern of variation of this biomarker [28].

Additionally, we found that mean red blood cell, white blood cell, and platelet counts, and hematocrit values were significantly lower in COVID-19 patients when compared with the control population. As above mentioned, the random forest classifier identifies the hematocrit value as a variable showing the most significance in distinguishing the LC from the nLC patients. This could be explained by several factors. In addition to its damaging effects on alveolar epithelial and endothelial cells, an excess of ROS may affect membrane lipids, integrin, and cytoplasmatic proteins in various circulating cells. Excess of ROS can cause oxidation of polyunsaturated fatty acid in red and white blood cells, bringing about a profound modification of the membrane lipid lateral and transversal distribution and organization at the nanoscale level, resulting in biophysical and biomechanical changes in cells membranes [47]. These biomechanical changes could be responsible for blood clots, identified as the signature of severe illness and, also, for microclots, identified by Pretorious et al. [48] even in LC patients, and potentially related to many chronic symptoms, such as cognitive difficulties, respiratory problems, and dysautonomia. In this light, lipid peroxidation could be a potential pathogenetic mechanism linking together immunological dysfunctions, inflammatory abnormal responses, and blood clots involved in the development of the long-COVID syndrome. However, further studies are needed to better clarify the biological importance of these results.

In our LC patients, more than twenty different types of symptoms were reported, and the more frequent were asthenia, insomnia, paresthesia, and concentration difficulties. In line with previous reports, long-term neurological and psychological complications seem to be not dependent on the severity of the acute disease, making early recognition more difficult due to the inconsistencies of neurological complications in a range of patients [49]. Many people who experience neurologic symptoms that linger after acute COVID-19 are less than 50 years old and were healthy and active prior to infection. Notably, the majority were never hospitalized during their acute COVID-19 illness, reflecting a mild initial disease [50].

The susceptibility of hippocampal and brainstem cells to coronaviruses may increase the risk of COVID-19 patients developing impaired cognitive function later [51]. Additionally, SARS-CoV-2 has been shown to cause nervous system alterations not only through direct infection pathways (both neuronal and circulatory) but also through secondary hypoxia, immune-mediated tissue damage, procoagulative and prothrombotic states [52]. Many of the symptoms experienced by individuals with long COVID are similar to those of myalgic encephalomyelitis/chronic fatigue syndrome (ME/CFS), which is also considered to be a postinfectious syndrome caused by a variety of infectious agents. Evidence shows that subjects with long COVID and ME/CFC share redox imbalance, systemic inflammation, impaired production of ATP, and abnormalities that have a bidirectional connection. In both conditions, symptoms such as brain fog and asthenia may be generated by mitochondrial damage, increased ROS production promoting inflammation, and chronic endothelial dysfunction, culminating in vicious cycles of neuroinflammation, reduced cerebral perfusion due to autonomic dysfunction, and autoantibodies directed to neural target [53].

This study has several strengths that should be addressed. Most importantly, it provides convincing evidence of a significant increase in serum MDA concentrations in non-hospitalized patients four months after the negativization of previous mild COVID-19. This cohort of non-hospitalized subjects is unique and provides new evidence on the presence of oxidative stress in long COVID, complementing previous studies [36]. At the same time, the small sample size possibly prevented the demonstration of potential association (e.g., between MDA and specific symptoms) with smaller effect. Moreover, oxidative stress biomarkers were compared only at one time. Larger follow-up studies with serial sampling should validate our results. Finally, this study was conducted in a single center and should be evaluated in a multicenter design to validate the potential role of MDA and lipid peroxidation in predicting the risk of the long-COVID syndrome. Nevertheless, our results may help to suggest the future performance of a larger-scale study. Finally, our findings, can provide important evidence helpful in the management of the occupational settings where workers with previous COVID-19 are present, considering the possible co-exposure to other occupational risk factors that can cause an additive effect of oxidative stress, increasing the risk of long COVID. This is very important in order to suggest the most appropriate preventive measures for optimal return to work activities even in individuals with a mild or moderate infection.

## 4. Materials and Methods

### 4.1. Study Population

This cross-sectional study was conducted among the employees of an Italian university (Bari, Apulia). A total of 80 workers with a previous COVID-19 infection (COVID-19 patients) were enrolled between January and June 2021, while 47 healthy employees, recruited in January 2020 in a previous study and therefore not reporting history of COVID-19 infection, were considered as a control group. The workers recruited in the study were subdivided into the following categories: full professor, associate professor, senior researcher, and tenure-tracked researcher, technical clerk, or administrative employee.

COVID-19 patients were recruited during the occupational health surveillance visit performed before returning to work, immediately after the SARS-CoV-2 infection test became negative. All the enrolled patients had been diagnosed at the Laboratory of Molecular Epidemiology and Public Health of the Hygiene Unit of the Policlinico Hospital of Bari, the Regional Reference Laboratory for COVID-19. This laboratory processes nasopharyngeal swabs collected from both hospitalized and non-hospitalized patients, covering 60% of the population of the province of Bari (1,230,205 inhabitants). Nasopharyngeal swabs (UTM, FLOQ Swabs TM, Copan Italia, Brescia, Italy) were subjected to molecular testing using a commercial, real-time PCR assay, as previously described [54]. At the time of the visit, the subjects reported the symptoms experienced during the acute phase of illness and the duration of these symptoms to appropriately trained medical personnel. Cases were defined according to the National Institutes of Health (NIH) clinical staging of COVID-19 [55]. In particular, COVID-19 patients were classified as “asymptomatic” if no signs or symptoms of COVID-19 were present; “mild” if symptoms, such as fever, cough, sore throat, malaise, headache, muscle pain, nausea, vomiting, diarrhea, or loss of taste and smell were present, but there was no evidence of shortness of breath, dyspnea, or abnormal chest imaging; “moderate” if there was evidence of lower respiratory disease during clinical assessment, but hospitalization was not required; “severe” if SpO_2_ was <94%, respiratory rate was >30 breaths/min, or chest imaging showed >50% lung infiltrates, with signs and symptoms of respiratory disease severe enough to require hospitalization; and “critical” if respiratory failure, septic shock, and/or multiple organ dysfunction had occurred, with the patient requiring admission to the intensive care unit. The patients were subsequently re-contacted four months after the first visit to conduct a new clinical assessment on a voluntary basis. At that time, the workers underwent venous blood sampling, and specific questionnaires were administered to them. Among COVID-19 patients, the exclusion criteria were to have been hospitalized with a moderate infection, defined as subjects with clinical signs of pneumonia (fever, cough, dyspnea, and fast breathing) or to have been affected by severe pneumonia during the COVID-19, such as SpO_2_ ≤ 90% on room air or to have been vaccinated against SARS-CoV-2. Further exclusion criteria for both COVID-19 patients and controls were a previous diagnosis of other chronic viral infections, such as HIV, HCV, HBV (in particular, patients were asked if they had ever been tested for such viruses); a solid organ or hematological transplantation in the past five years; and reports of special regimen or taking antioxidant supplements, such as vitamin C, vitamin E, Coenzyme Q10, and selenium. All the workers were clinically evaluated to exclude other possible causes related to the persistence of their clinical symptoms. The principles of ICH Good Clinical Practice, the ‘Declaration of Helsinki’ and national and international ethical guidelines were strictly followed during this study. The study was approved by the research Ethics Committee of the University Hospital of Bari (no. 7662) and all the subjects signed the informed consent form.

### 4.2. Questionnaires

All participants were interviewed face-to-face by trained physicians and asked to complete a general and symptom questionnaire.

Data on COVID-19 patients’ demographic characteristics and regular medication use, symptoms at COVID-19 diagnosis, and the type and number of comorbidities, as described in Stufano et al. [56], were collected during the first clinical visit. At that time, we additionally collected data on demographic characteristics (age, sex, education, and cigarette smoking), clinical characteristics (symptom onset time and possible chest images), laboratory test results, and COVID-19 treatment (corticosteroids, intravenous immunoglobulin, antibiotics, and antivirals). At the follow-up visit, data regarding patients’ symptoms four months after the first negative PCR test, including fever, cough, dyspnea, ageusia, anosmia, diarrhea, arthralgia, myalgia, chest pain, sore throat, headache, and perception of reduced tolerance to physical activity compared with before they contracted COVID-19, were further collected. Moreover, participants were asked to report newly occurring and persistent symptoms, or any symptoms worse than before COVID-19 development. In accordance with NICE and WHO definitions [3,4], long-COVID (LC) patients were identified among the group of COVID-19 patients, selecting those who reported the persistence of at least one symptom after infection, or the appearance of symptoms that could not be explained by other causes and were plausibly related to COVID-19 infection.

### 4.3. Blood Sample Collection and Biochemical Assays

Venous blood samples (10 mL) were collected on the same day of the administration of the questionnaires. Peripheral blood was collected in vacutainers without additives containing separating gel and kept at room temperature for 30 min to coagulate, then centrifuged at 1600 rpm for 10 min at 4 °C. The tubes then remained at rest for 1 h in a vertical position; the serum was aliquoted and stored at −70 °C until biochemical analyses were conducted.

After blood samples collection, routine laboratory tests were performed on all participants, including hemoglobin (Hb), white blood cell count (WBC), and platelet counts, by an automated hematology analyzer (Sysmex XE-2100, Sysmex Corporation, Kobe, Japan), whereas a photometric method was used to measure serum creatinine (Roche Modular, Roche Diagnostics, Mannheim, Germany).

A chemiluminescent immunoassay that detects IgG against the nucleocapsid protein of SARS-CoV-2 was performed on the same serum samples (Abbott SARS-CoV-2 IgG assay on the Abbott Architect i4000SR; Abbott Diagnostics, Chicago, IL, USA) A signal/cut-off (S/CO) ratio ≥ 1.4 was interpreted as reactive, and a S/CO ratio < 1.4 as nonreactive, as per the manufacturer’s instructions.

Serum(LPO) was analyzed by the thiobarbituric acid-reactive substance (TBARS) assay (Cayman Chemical TBARS Assay Kit, Minneapolis, MN, USA) as described in [57] with some modifications. The TBARS assay detects the level of MDA, the major lipid oxidation product, and some minor related compounds. Serum samples were immediately stored, then 50 µL were incubated in 50 µL of sodium dodecylsulphate (SDS) lysis solution to denature the proteins. Then, TBA was added, and the samples were incubated for 60 min at 100 °C. The tubes were cooled in ice for 10 min to stop the reaction and then centrifuged at 1600× *g* for 10 min at 4 °C. The supernatants were recovered, and optical density was measured by a multilabel plate reader at 530 nm (Infinite 200 Pro, Tecan, Italy).

Total hydroperoxide (TH) production was measured by a d-ROMs Kit (Diacron Srl, Grosseto, Italy), as described in [58] with some modifications. This method is based on Fenton’s reaction, where the hydroperoxides present in the sample react with iron generating alkyl (R-O) and peroxyl (R-OO) radicals. The intensity of the color change in the sample is directly proportional to the concentration of reactive oxygen metabolites (ROMs) in the sample. Briefly, 6 μL of the sample was mixed with 6 μL of reagent 1 and 600 μL of reagent 2. After an incubation of 3 min at 37 °C, the optical density was measured by using a multilabel plate reader at 505 nm (Infinite 200 Pro, Tecan, Italy). The measures were repeated after another 2 min of incubation, and the change in absorbance (∆A) was calculated. Finally, the results were expressed in conventional arbitrary units, called Carratelli (Carr) units. The value of 1 Carr unit corresponds to a concentration of 0.08 mg/dl of hydrogen peroxide (H_2_O_2_). A hydrogen peroxide calibration curve was made using titrated H_2_O_2_ solutions.

### 4.4. Statistical Analysis

Shapiro–Wilk test and graphical evaluations of each variable were performed to demonstrate the correspondence with the normal distribution. Student *t*-test or Mann–Whitney U test was performed to assess comparisons between the two groups in terms of continuous variables, while differences between more than two groups were studied through one-way analysis of variance (ANOVA) followed by Tukey post hoc test or Kruskal–Wallis test where necessary. Pearson χ^2^ test was used for comparisons in terms of categorical variables. The ROC curves were used to determine optimum cut-off levels of nonoxidative damage indexes and *p* values lower than 0.05 were regarded as significant.

To assess the interplay among some clinical and demographic variables and to identify internally homogeneous subgroups of patients with different risks of developing long COVID, a random forest classifier was applied. A random forest is a set of decision trees. Consequently, we were able to interrogate this collection of trees to identify the features that have the highest predictive value (viz., those features that frequently appear near the top of the decision tree). To reduce overfitting and maintain a conservative model, threefold cross-validation with a random forest of 10 trees and a maximum depth of 3 was used [59]. All analyses were performed using SPSS version 23 (SPSS Inc., Chicago, IL, USA).

## 5. Conclusions

In summary, our findings deepen our knowledge of oxidative stress status in long COVID, demonstrating higher lipid peroxidation in subjects with persisting symptoms after SARS-CoV-2 infection. Moreover, we demonstrated that non-hospitalized individuals with previous COVID-19 are also markedly affected by systemic oxidative stress in comparison with healthy controls. It is important to underline that slowed cognitive processing speed and memory impairment, fatigability, and deficits of concentration referred to in our LC patients could interfere with daily and work-related functioning. We believe that workers may benefit from early neuropsychological and oxidative damage assessments in order to evaluate the degree of impairment following COVID-19 and its impact on their ability to return to work.

## Figures and Tables

**Figure 1 ijms-24-07445-f001:**
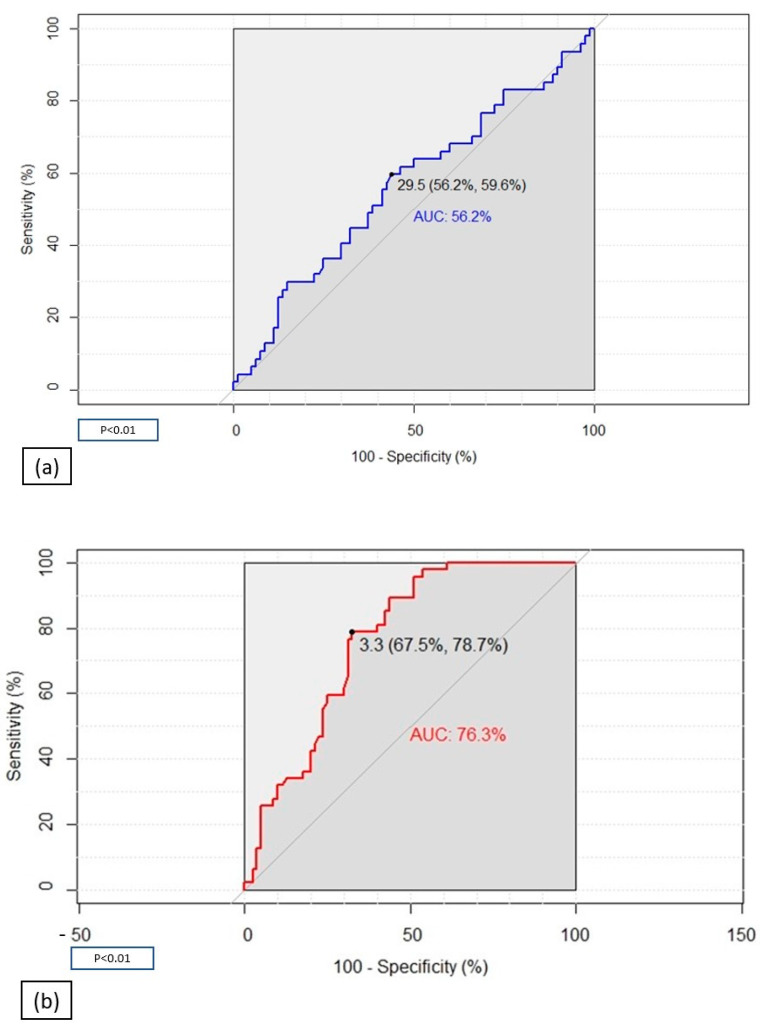
Receiver–operating characteristic (ROC) analyses assessing the ability of H_2_O_2_ (**a**) and MDA (**b**) to discriminate COVID-19 patients from healthy controls.

**Figure 2 ijms-24-07445-f002:**
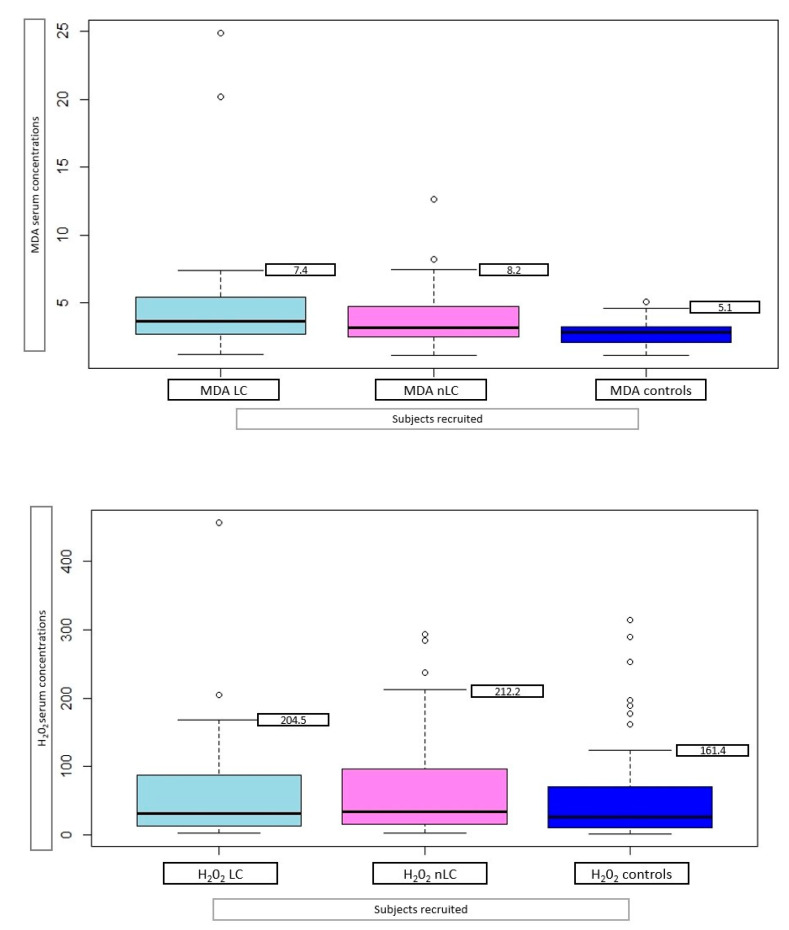
Serum mean MDA (µm) and H_2_O_2_ (mg/dL), with 95% family-wise confidence level, in 34 long-COVID (LC) patients, 46 non-long-COVID (nLC), and 47 controls.

**Figure 3 ijms-24-07445-f003:**
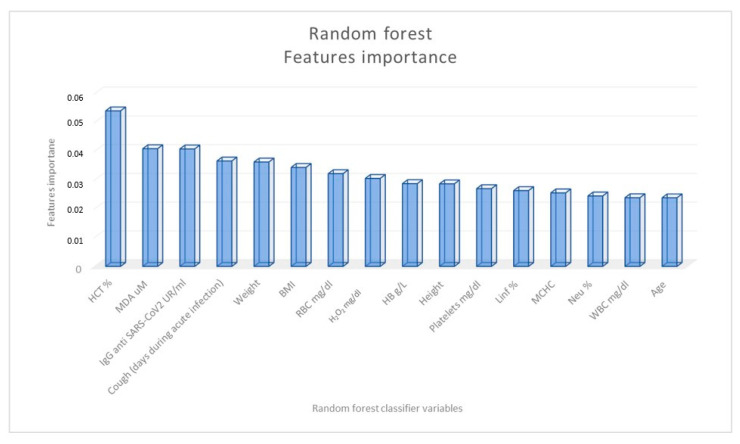
Importance of the random forest classification variables in distinguishing COVID-19 patients according to the presence (long-COVID patients) or not (non-long-COVID patients) of at least one symptom four months after negativization.

**Table 1 ijms-24-07445-t001:** General characteristics and clinical parameters of COVID-19 patients and controls.

General Characteristics	COVID-19 Patients (N 80)			Controls (N 47)		
	N (%)	Mean ± SD	Range	N (%)	Mean ± SD	Range
Age (years)		50.6 ± 9.3	28–68		53.8 ± 12.6	31–73
BMI (Kg/m^2^)		25.8 ± 4.5	17.6–52.2		24.0 ± 3.3	18.1–32.4
Gender						
Men	56 (70)			33 (70)		
Women	24 (30)			14 (30)		
Smoking habits						
Smokers	25 (31)			16 (34)		
Non-smokers	55 (69)			31 (66)		
Alcohol consumption (U/week) ^b^		2.1 ± 5.5	1–7		3.6 ± 4.6	0–14
RBC (mg/dL) ^a^		4.9 ± 0.5	3.5–7.1		5.2 ± 0.5	4.4–7.6
WBC (mg/dL) ^aa^		6.5 ± 1.3	4.1–11.1		7.3 ± 1.4	5.4–13.9
Hb (g/dL)		13.9 ± 1.2	10.8–16.4		13.9 ± 0.7	12.1–15.8
HCT (%) ^a^		41.5 ± 4.5	15.8–48.0		42.8 ± 0.2	38.7–48.7
MCV (fl/cell) ^a^		86.4 ± 4.8	61.1–97.0		83.8 ± 5.0	63.0–91.0
MCHC (pg/dL) ^a^		29.2 ± 2.3	19.3–34.7		27.0 ± 1.9	18.9–29.1
Neutrophils (%)		55.0 ± 8.4	17–72		56.6 ± 6.8	44–78
Lymphocytes (%)		34.3 ± 8.1	16–76		32.5 ± 5.4	17–44
Monocytes (%)		8.0 ± 1.8	2–12		7.7 ± 2.1	2–14
Eosinophils (%)		2.4 ± 1.6	0–8		2.4 ± 1.4	1–8
Basophils (%)		0.3 ± 0.5	0–2		0.6 ± 0.6	0–2
Platelets (mg/dL) ^a^		239.0 ± 52.2	97–398		251.3 ± 36.3	116–347
H_2_O_2_ (mg/dL)		65.4 ± 80.0	1.8–465.5		59.2 ± 78.7	1.5–313.0
MDA (µm) ^a^		4.9 ± 3.5	1.2–24.9		2.8 ± 0.9	1.2–5.1

RBC: red blood cells; WBC: white blood cells; Hb: hemoglobin; HCT: hematocrit value; MCV: mean corpuscular volume; MCHC: mean corpuscular hemoglobin concentration. ^a^ = *p* ≤ 0.001; ^b^ = *p* ≤ 0.05.

**Table 2 ijms-24-07445-t002:** General characteristics and clinical parameters of COVID-19 patients subdivided according to the presence (long-COVID patients) or not (non-long-COVID patients) of at least one symptom four months after negativization.

General Characteristics	Long-COVID Patients (N. 34)			Non-Long-COVID Patients (N. 46)		
	N (%)	Mean ± SD	Range	N (%)	Mean ± SD	Range
Age (years)		50.9 ± 9.4	28–66		50.3 ± 0.7	30–68
Body Mass Index (Kg/m^2^)		25.3 ± 4.1	17.6–34.6		26.1 ± 1.8	21.6–52.2
Gender ^a^						
Men	15 (44)			41 (89)		
Women	19 (56)			6 (11)		
Smoking habits						
Smokers	11 (32)			14 (30)		
Nonsmokers	23 (68)			32 (70)		
Alcohol consumption(U/week)		2.1 ± 2.7	1–7		2.2 ± 3.5	0–6
Comorbidities						
Cardiovascular diseases	6 (17.6)			10 (21.7)		
Respiratory diseases	6 (17.6)			4 (8.6)		
Autoimmune diseases	4 (11.7)			1 (2.1)		
Neoplasms	2 (5.8)			3 (6.5)		
RBC (mg/dL)		4.7 ± 0.4	3.7–5.4		4.9 ± 0.1	3.5–7.1
WBC (mg/dL)		6.5 ± 1.2	4.1–9.7		6.5 ± 1.4	4.6–11.1
Hb (g/dL)		13.5 ± 1.3	10.8–16.1		14.1 ± 1.6	11.2–16.4
HCT (%) ^a^		39.7 ± 5.6	15.9–46.8		42.7 ± 1.4	33.0–48.0
MCV (fl/cell)		86.3 ± 4.2	75.0–97.0		86.4 ± 2.5	61.0–96.8
MCHC (pg/dL)		28.9 ± 1.4	25.5–32.3		29.3 ± 3.5	19.3–34.7
Platelets (mg/dL)		241.0 ± 60.9	97–398		238.2 ± 62.1	152–347
IgG Anti-SARS-CoV-2 (UR/mL) ^a^		68.1 ± 72.7	0.0–223.0		26.6 ± 9.0	0.0–246.0
H_2_O_2_ (mg/dL)		61.9 ± 66.1	1.7–456.4		67.9 ± 70.6	2.6–294.0
MDA (µm) ^a^		5.7 ± 5.9	1.2–24.8		3.8 ± 2.4	1.1–12.6
Therapy						
Paracetamol	11 (32.3)			12 (26.1)		
Corticosteroids	13 (38.2)			12 (26.1)		
Oxygen therapy	4 (11.7)			0 (0.0)		
Heparin	6 (17.6)			2 (4.3)		
Antiviral drugs	0 (0.0)			0 (0.0)		

RBC: red blood cells; WBC: white blood cells; Hb: hemoglobin; HCT: hematocrit value; MCV: mean corpuscular volume; MCHC: mean corpuscular hemoglobin concentration. ^a^
*p* ≤ 0.001.

**Table 3 ijms-24-07445-t003:** Clinical characteristics of COVID-19 in patients subdivided according to the presence (long-COVID patients) or not (non-long-COVID patients) of at least one symptom four months after negativization.

Prevalence and Duration of Symptoms during COVID-19 Infection (Days)	Long-COVID Patients (N. 34)			Non-Long-COVID Patients (N. 46)		
	N(%)	Mean	Range	N (%)	Mean	Range
Duration of COVID-19 infection (days)		23.3 ± 1.4	10–64		22.6 ± 1.3	2–52
Presence of symptoms during infection	32 (94.1)			41 (89.1)		
Fever	24 (70)	0.7 ± 5.6	0–24	27 (58)	2.4 ± 7	0–20
Dyspnea	9 (26)	4.7 ± 0.7	0–65	10 (21)	9.2 ± 11.3	0–160
Cough ^a^	20 (58)	9.2 ± 7.7	0–39	16 (34)	3.1 ± 6.3	0–24
Myalgia	21 (61)	5.9 ± 7.0	0–45	21 (45)	11.6 ± 22.6	0–151
Pharyngodynia	6 (17)	0.9 ± 0.0	0–10	7 (15)	1.3 ± 11.3	0–38
Ageusia	17 (50)	7.6 ± 1.4	0–41	21 (46)	8.7 ± 5.6	0–115
Anosmia	19 (55)	13.5 ± 0.0	0–141	24 (52)	8.1 ± 5.6	0–115
Diarrhea	8 (23)	0.7 ± 1.4	0–10	8 (17)	1.2 ± 0.0	0–22
Headache ^b^	11 (32)	2.5 ± 0.0	0–19	10 (22)	0.7 ± 0.0	0–7
Dermatitis	1 (3)	0.2 ± 0.0	0–7	2 (5)	2.3 ± 0.0	0–103
Asthenia	8 (23)	4.0 ± 7.7	0–31	4 (9)	2.1 ± 0.7	0–48

^a^*p* ≤ 0.001; ^b^
*p* ≤ 0.05.

## Data Availability

The data presented in this study are available on request from the corresponding author.

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
