# Peer review of "Oxidative Damage and Post-COVID Syndrome: A Cross-Sectional Study in a Cohort of Italian Workers"

_ijms, 2023, doi:10.3390/ijms24087445_

Round 1

Reviewer 1 Report

Assessing the antioxidant status in post COVID syndrome is an interesting field. Several points in the manuscript are not clarified in detail:

·       Table 1.  Please add abbreviations: MDA, RBC, WBC…etc.

·       row 110 Covid.19 instead of COVID-19

·       row 116-119: The definition should be transfered to Methods

·       Figure 2. Please indicate significance within the Figure not only in the text

·       row 108-109: The ROC curve of MDA otherwise was characterized by high specificity and good sen-sibility (78.7% and 67.5%, respectively) identifying MDA as a good predictive index of 109 COVID.19 patients, with a cut-off value of 3.3 μm/dl, and AUC of 76.3%.

For 78.7% percent, taliking about high sensitivity feels a bit exaggerated.

·       row 128-129 and row 238-240: In addition, this group had significantly higher IgG values and 128 lower mean hematocrit values than the AsPC patients (Table 2). According to our 238 results, showing significant higher mean SARS-CoV-2 IgG titers in LC patients, immuno-239 logical dysfunctions have been detected in different studies including Long Covid sub-240 jects.

This is a very important statement, it would be important to cite one or two publications in thsi field in the discussion section. Please add: Molnar, T., Varnai, R., Schranz, D., Zavori, L., Peterfi, Z., Sipos, D., TÅ‘kés-Füzesi, M., Illes, Z., Buki, A., & Csecsei, P. (2021). Severe Fatigue and Memory Impairment Are Associated with Lower Serum Level of Anti-SARS-CoV-2 Antibodies in Patients with Post-COVID Symptoms. Journal of clinical medicine10(19), 4337. https://doi.org/10.3390/jcm10194337

Fogh, K., Larsen, T. G., Hansen, C. B., Hasselbalch, R. B., Eriksen, A. R. R., Bundgaard, H., Frikke-Schmidt, R., Hilsted, L. M., Østergaard, L., Johansen, I. S., Hageman, I., Garred, P., & Iversen, K. (2022). Self-Reported Long COVID and Its Association with the Presence of SARS-CoV-2 Antibodies in a Danish Cohort up to 12 Months after Infection. Microbiology spectrum10(6), e0253722. https://doi.org/10.1128/spectrum.02537-22

·       row 351-352: Dose this mean that all the participants had HBV, HCV, HIV test sor were they just assumed to be negative if they had no symptoms?

·       In Table 2 i cannot see any infos about antivirals during acute disease, however at Methods the Authors stated that they ask volunteers about it. Its a crucial point cause antivirals (remdesevir, favipiravir) can modify redox levels: Ref: Villani, R., Bellanti, F., Cavallone, F., Di Bello, G., Tamborra, R., Bukke Vidyasagar, N., Moola, A., & Serviddio, G. (2020). Direct-acting antivirals restore systemic redox homeostasis in chronic HCV patients. Free radical biology & medicine156, 200–206. https://doi.org/10.1016/j.freeradbiomed.2020.06.007

Proskurnina, E. V., Izmailov, D. Y., Sozarukova, M. M., Zhuravleva, T. A., Leneva, I. A., & Poromov, A. A. (2020). Antioxidant Potential of Antiviral Drug Umifenovir. Molecules (Basel, Switzerland)25(7), 1577. https://doi.org/10.3390/molecules25071577

·       Why were moderate and severe COVID-19 patients excluded from the study?

Overall, the work makes important findings. After clarifying the points indicated above, I am waiting for the corrected version.

Author Response

REVIEWER #1: Assessing the antioxidant status in post COVID syndrome is an interesting field. Several points in the manuscript are not clarified in detail:

-Table 1.  Please add abbreviations: MDA, RBC, WBC…etc.

Authors’ response:We thank the reviewer for the suggestion, and we modify the text accordingly.

-Row 110 Covid.19 instead of COVID-19

Authors’ response:We thank the reviewer for the recommendation, however we decided to use the term 'COVID-19' as it refers to the official nomenclature of the disease in accordance with the World Health Organisation definition (see the link:  https://covid19.who.int/)

-Row 116-119: The definition should be transferred to Methods

Authors’ response:The text has been changed in accordance with the reviewer's suggestion. Thank you for your comment.

-Figure 2. Please indicate significance within the Figure not only in the text

Authors’ response:The figure has been modified according to the reviewer observation. Thanks for the suggestion.

-Row 108-109: The ROC curve of MDA otherwise was characterized by high specificity and good sen-sibility (78.7% and 67.5%, respectively) identifying MDA as a good predictive index of 109 COVID.19 patients, with a cut-off value of 3.3 μm/dl, and AUC of 76.3%. For 78.7% percent, talking about high sensitivity feels a bit exaggerated.

Authors’ response:We thank the author for his comment and have modified the text in accordance with his suggestion.

-Row 128-129 and row 238-240: In addition, this group had significantly higher IgG values and 128 lower mean hematocrit values than the AsPC patients (Table 2). According to our 238 results, showing significant higher mean SARS-CoV-2 IgG titers in LC patients, immuno-239 logical dysfunctions have been detected in different studies including Long Covid sub-240 jects.

This is a very important statement; it would be important to cite one or two publications in this field in the discussion section. Please add:Molnar, T., Varnai, R., Schranz, D., Zavori, L., Peterfi, Z., Sipos, D., TÅ‘kés-Füzesi, M., Illes, Z., Buki, A., &Csecsei, P. (2021). Severe Fatigue and Memory

Impairment Are Associated with Lower Serum Level of Anti-SARS-CoV-2 Antibodies in Patients with Post-COVID Symptoms. Journal of clinical medicine, 10(19), 4337. https://doi.org/10.3390/jcm10194337

Fogh, K., Larsen, T. G., Hansen, C. B., Hasselbalch, R. B., Eriksen, A. R. R., Bundgaard, H., Frikke-Schmidt, R., Hilsted, L. M., Østergaard, L., Johansen, I. S., Hageman, I., Garred, P., & Iversen, K. (2022). Self-Reported Long COVID and Its Association with the Presence of SARS-CoV-2 Antibodies in a Danish Cohort up to 12 Months after Infection. Microbiology spectrum, 10(6), e0253722. https://doi.org/10.1128/spectrum.02537-22

Authors’ response:Thanking the reviewer for the suggestion, we have added the cited references to the text.

-Row 351-352: Does this mean that all the participants had HBV, HCV, HIV test or were they just assumed to be negative if they had no symptoms?

Authors’ response: We thank the reviewer for his comments that enable us to improve the clarity of our manuscript.The recruited subjects were administered a questionnaire, in which they were asked whether they had ever been tested for these viruses, and the result, if any. None of the subjects reported symptoms compatible with these diseases, but only a percentage of them (38%) had been tested for the above-mentioned viruses. We have reported this information in the methods section.

-In Table 2 I cannot see any info about antivirals during acute disease, however at Methods the Authors stated that they ask volunteers about it. It’s a crucial point cause antivirals (remdesevir, favipiravir) can modify redox levels: Ref: Villani, R., Bellanti, F., Cavallone, F., Di Bello, G., Tamborra, R., BukkeVidyasagar, N., Moola, A., &Serviddio, G. (2020). Direct-acting antivirals restore systemic redox homeostasis in chronic HCV patients. Free radical biology & medicine, 156, 200–206. https://doi.org/10.1016/j.freeradbiomed.2020.06.007; Proskurnina, E. V., Izmailov, D. Y., Sozarukova, M. M., Zhuravleva, T. A., Leneva, I. A., &Poromov, A. A. (2020). Antioxidant Potential of Antiviral Drug Umifenovir. Molecules (Basel, Switzerland), 25(7), 1577. https://doi.org/10.3390/molecules25071577.

Authors’ response: We thank the reviewer for the helpful remark, none of the recruited subjects received antiviral therapy, we have reported the result in table 2 as requested.

-Why were moderate and severe COVID-19 patients excluded from the study?

Authors’ response: In the present study, we decided to evaluate the possible long-term effects of COVID-19 infection, and possible alterations in the oxidative status of the same subjects, exclusively in mildly infected subjects, to identify possible risk factors for the development of post-Covid syndromes also in this cohort type as it has been little studied from this point of view.

-Overall, the work makes important findings. After clarifying the points indicated above, I am waiting for the corrected version.

Authors’ response: We thank the reviewer for the positive opinion of our work and for the valuable suggestions that enabled us to improve the quality of our manuscript.

Reviewer 2 Report

 Clarity of presentation,  soundness of the methods, thorough description of imitations and adequacy of references add to the overall value of the manuscript. All in all, it is worth publishing in the present form. It is a very interesting work, although the sample size is not very large. The methods are at the end of the job. Modify it.

Author Response

REVIEWER #2: Clarity of presentation,  soundness of the methods, thorough description of imitations and adequacy of references add to the overall value of the manuscript. All in all, it is worth publishing in the present form. It is a very interesting work, although the sample size is not very large. The methods are at the end of the job.Modify it.

Authors’ response: We thank the reviewer for the comments to our study. We put the method on the end because it was required from the editorial office of the journal

Reviewer 3 Report

The manuscript by Stufano and Isgrò et al. evaluates the relationship between changes in oxidative status and the persistence of long COVID symptoms in healthcare workers with previous mild COVID-19. This study is interesting, easy to follow and well written, and shows novelty because is the second one observing that oxidative damage persists in subjects with previous COVID-19 infection and that it might be associated with long COVID. However, there are some flaws that the authors should address before having this work considered for publication in any journal.

Major comments:

-        Regarding the study cohort, the percentage of long COVID seems too high, currently it usually is between 10-20% of COVID infections and in this study the long COVID participants represent more than 40% of the COVID+ evaluated participants. This might be causing an overestimation of long COVID and considering cases that are not “real” long COVID cases.

-        In this regard, the authors only considered as long COVID people with one persistent symptom 4 months after acute COVID-19. They should follow the WHO definition of long COVID to define their cohort, to be more accurate in the definition of long COVID.

-        The references are not accurate in all the citations and most of them do not correspond to what the authors state in the text. The authors should revise the bibliography to cite the references correctly accordingly to what they state.

-        It is not clearly specified whether the participants included have been clinically diagnosed with long COVID. The authors should add that information to the text in the methods section.

-         In lines 102-103 the authors specify that MDA levels are higher in controls than in COVID-19 patients, when in the table the values are higher in COVID-19 patients and in lines 167-168 they stated the contrary. Authors should revise this information because it is the main conclusion of the manuscript.

Minor comments:

-         The authors should change “asymptomatic post-Covid (AsPC)” and maybe use “recovered patients” or “non-Long COVID”, because the first name might be misleading.

-         They authors claim that their work is the first one associating oxidative damage with long COVID, but in October 2023 an article about this was published (PMID: 36280755 PMCID: PMC9589528 DOI: 10.1038/s41380-022-01836-9). Therefore, the manuscript by Stufano and Isgrò et al. might be the second one, but not the first.

-        In the figures the authors should specify the title/units of the axis.

-        As statistical analysis, the authors confirm that they performed normality tests to verify normal distribution. However, some of the data they analyze does not look like they follow a normal distribution and considering the low sample size, they should also run non-parametric tests to verify their results.

-        In figure 2, to facilitate the interpretation, sample size should be added to the figure or the figure legend. Also, it might help to represent individual values connected with lines on top of the box plots to follow the individual dynamic of the values.

-        In figure 3, it might facilitate the interpretation and reading if the symptoms are ordered from high to low % or viceversa. Also, the authors should specify if these symptoms correspond to long COVID or asymptomatic subcohorts. This information is also repeating the information on table 3. Authors might decide which representation they prefer to avoid repetition.

-        In tables the acronyms should be specified in the table legend.

-        In the random forest seems that the most significant variable related to COVID-19 patients is HCT (%), although looking at the mean in table 1, the difference between long COVID and asymptomatic postCOVID patients does not seem too high (3%). What do the authors think the biological relevance of this observation and of the results of the random forest might be?

-        When comparing long COVID with asymptomatic postCOVID patients, statistically significant differences were observed regarding the gender. Do the authors know whether any of the other parameters compared between these two subcohorts might be associated with gender/sex? In another words, might gender/sex be a confounding element of the analysis?

-        In table 2, exact levels of MDA and H2O2 should be included.

Author Response

REVIEWER #3: The manuscript by Stufano and Isgrò et al. evaluates the relationship between changes in oxidative status and the persistence of long COVID symptoms in healthcare workers with previous mild COVID-19. This study is interesting, easy to follow and well written, and shows novelty because is the second one observing that oxidative damage persists in subjects with previous COVID-19 infection and that it might be associated with long COVID. However, there are some flaws that the authors should address before having this work considered for publication in any journal.

Major comments:

-Regarding the study cohort, the percentage of long COVID seems too high, currently it usually is between 10-20% of COVID infections and in this study the long COVID participants represent more than 40% of the COVID+ evaluated participants. This might be causing an overestimation of long COVID and considering cases that are not “real” long COVID cases.

-In this regard, the authors only considered as long COVID people with one persistent symptom 4 months after acute COVID-19. They should follow the WHO definition of long COVID to define their cohort, to be more accurate in the definition of long COVID.

Authors’ response: We thank the reviewer for the comment, which allowed us to better clarify the results of our study. In our study, we chose to use the definition provided by NICE as this institute provided the first official definition of Long Covid, based on literature evidences and focused on a clinical approach aimed at protecting patients with this condition. However, following the reviewer's appropriate consideration, we compared our selection criterion with the WHO definition of Long Covid, i.e. 'the continuation or development of new symptoms 3 months after the initial SARS-CoV-2 infection, with these symptoms lasting for at least 2 months without other explanation'. When analysing this definition, we did not find any substantial differences with the definition provided by NICE, as this definition also does not include the number of symptoms required to make a diagnosis of Long Covid, and at the same time the temporal recruitment criteria were met, as in our study sample, subjects identified as having Long Covid syndrome were recruited three months after negativity. Furthermore, although our identification criterion was linked to the presence of at least one persistent symptom four months after negativity, in our study sample, none of the subjects identified as having Long Covid syndrome were recruited three months after negativity. Furthermore, in our study sample, none of the subjects reported the persistence of a single symptom, but all reported the persistence of at least two symptoms. Following the reviewer's suggestion, we reported the average number of symptoms and the range reported by workers with Long Covid in Table 3. Following the reviewer's suggestion, we decided to refer to the WHO and NICE definition of Long Covid. Furthermore, although the WHO estimated a prevalence of Long Covid of about 20%, reports on European populations, such as the ECDC report on Long Covid, reported a prevalence of any post-COVID-19 symptoms among EU-recruited cohorts of 50.6% (European Centre for Disease Prevention and Control. (Prevalence of post COVID-19 symptoms: A systematic review and meta-analysis of cohort study data stratified by recruitment setting. 27 October 2022. ECDC: Stockholm; 2022). Following the reviewer's suggestion, we have added these different reports to the discussion.

-The references are not accurate in all the citations and most of them do not correspond to what the authors state in the text. The authors should revise the bibliography to cite the references correctly accordingly to what they state.

Authors’ response: We thank the reviewer for the important observation, references were checked and modified accordingly.

-It is not clearly specified whether the participants included have been clinically diagnosed with long COVID. The authors should add that information to the text in the methods section.

Authors’ response: We thank the reviewer for the timely observation. The workers were recruited during health surveillance visits, so they underwent medical examinations both during the first visit and during the follow-up, and in both cases the administration of questionnaires and evaluation by additional examinations performed by the patients elsewhere made it possible to exclude other possible causes plausibly associated with their clinical symptoms. We have therefore specified this aspect in the methods, in accordance with the reviewer's suggestion (lines 360-361)

-In lines 102-103 the authors specify that MDA levels are higher in controls than in COVID-19 patients, when in the table the values are higher in COVID-19 patients and in lines 167-168 they stated the contrary. Authors should revise this information because it is the main conclusion of the manuscript.

Authors’ response: We thank the reviewer for the correction, we have modified the text in accordance with the suggestion.

Minor comments:

-The authors should change “asymptomatic post-Covid (AsPC)” and maybe use “recovered patients” or “non-Long COVID”, because the first name might be misleading.

Authors’ response: Following the appropriate suggestion of the reviewer, we have amended the text accordingly.

-They authors claim that their work is the first one associating oxidative damage with long COVID, but in October 2023 an article about this was published (PMID: 36280755 PMCID: PMC9589528 DOI: 10.1038/s41380-022-01836-9). Therefore, the manuscript by Stufano and Isgrò et al. might be the second one, but not the first.

Authors’ response:We thank the reviewer for the suggestion, in the text the study mentioned had already been previously cited, we have also modified the text as recommended.

-In the figures the authors should specify the title/units of the axis.

-In figure 2, to facilitate the interpretation, sample size should be added to the figure or the figure legend. Also, it might help to represent individual values connected with lines on top of the box plots to follow the individual dynamic of the values.

-In figure 3, it might facilitate the interpretation and reading if the symptoms are ordered from high to low % or viceversa. Also, the authors should specify if these symptoms correspond to long COVID or asymptomatic subcohorts. This information is also repeating the information on table 3. Authors might decide which representation they prefer to avoid repetition.

Authors’ response: According to the properly suggestion of the reviewer, we removed the figure 3 from the text and modified the figures as requested.

-As statistical analysis, the authors confirm that they performed normality tests to verify normal distribution. However, some of the data they analyze does not look like they follow a normal distribution and considering the low sample size, they should also run non-parametric tests to verify their results.

Authors’ response: We thank the reviewer for the suggestion, accordingly, we run also non parametric test to verify our results and specify it in the method section.

-In tables the acronyms should be specified in the table legend.

Authors’ response: We thank the reviewer for the comment, the acronyms were specified in the table’s legends.

-In the random forest seems that the most significant variable related to COVID-19 patients is HCT (%), although looking at the mean in table 1, the difference between long COVID and asymptomatic postCOVID patients does not seem too high (3%). What do the authors think the biological relevance of this observation and of the results of the random forest might be?

Authors’ response: We thank the reviewer for the comment. The importance of the hematocrit values could be related to the lipid peroxidation effect mediated from MDA, however, because of the low

differences of values between Long covid and non-Long Covid we think that this evidence could be related to some other confounding factors. Further studies could be necessary to confirm the biological importance of this difference.

-When comparing long COVID with asymptomatic postCOVID patients, statistically significant differences were observed regarding the gender. Do the authors know whether any of the other parameters compared between these two subcohorts might be associated with gender/sex? In another words, might gender/sex be a confounding element of the analysis?

Authors’ response: According to the reviewer's suggestion, we compared the other parameters described according to gender, and no significant differences were found. We specified it in the text.

-In table 2, exact levels of MDA and H2O2 should be included.

Authors’ response: We modified the table according to the suggestion.

Reviewer 4 Report

In this cross-sectional study, Stufano and colleagues evaluate the relationship between changes in oxidative status and the persistence of Long COVID symptoms in workers with a previous mild COVID-19. The Authors enrolled 127 employees of an Italian University (80 with a previous COVID-19 infection, and 47 healthy subjects). Malondialdehyde (MDA) serum levels (MDA) and total hydroperoxide (TH) production were measured. A significant difference in mean serum MDA values was found between healthy controls and previously infected subjects. Mechanisms are discussed.

I think this is a well-executed and novel study and adds to our knowledge about the many manifestations of COVID-19 and long-COVID syndrome. 

Author Response

REVIEWER #4: In this cross-sectional study, Stufano and colleagues evaluate the relationship between changes in oxidative status and the persistence of Long COVID symptoms in workers with a previous mild COVID-19. The Authors enrolled 127 employees of an Italian University (80 with a previous COVID-19 infection, and 47 healthy subjects). Malondialdehyde (MDA) serum levels (MDA) and total hydroperoxide (TH) production were measured. A significant difference in mean serum MDA values was found between healthy controls and previously infected subjects. Mechanisms are discussed.I think this is a well-executed and novel study and adds to our knowledge about the many manifestations of COVID-19 and long-COVID syndrome.

Authors’ response: We really thank the reviewer for the positive comments to our work.

Round 2

Reviewer 1 Report

The corrections made are appropriate, the paper is suitable for publication.

Congratulations!

Reviewer 3 Report

I thank the authors for replying and modifying the manuscript accordingly to the reviewers' comments.